# In Vitro Immunological Cross-Reactivity of Thai Polyvalent and Monovalent Antivenoms with Asian Viper Venoms

**DOI:** 10.3390/toxins12120766

**Published:** 2020-12-03

**Authors:** Janeyuth Chaisakul, Muhamad Rusdi Ahmad Rusmili, Jaffer Alsolaiss, Laura-Oana Albulescu, Robert A. Harrison, Iekhsan Othman, Nicholas R. Casewell

**Affiliations:** 1Department of Pharmacology, Phramongkutklao College of Medicine, Bangkok 10400, Thailand; 2Kulliyyah of Pharmacy, International Islamic University Malaysia, Kuantan Campus, Bandar Indera Mahkota, Kuantan 25200, Pahang Darul Makmur, Malaysia; rusdirusmili@iium.edu.my; 3Centre for Snakebite Research & Interventions, Liverpool School of Tropical Medicine, Pembroke Place, Liverpool, Merseyside L3 5QA, UK; Jaffer.AlSolaiss@lstmed.ac.uk (J.A.); Laura-Oana.Albulescu@lstmed.ac.uk (L.-O.A.); Robert.Harrison@lstmed.ac.uk (R.A.H.); 4Jeffrey Cheah School of Medicine and Health Sciences, Monash University Sunway Campus, Bandar Sunway 46150, Malaysia; iekhsan.othman@monash.edu

**Keywords:** snakebite, venom, vipers, antivenom, cross-neutralization, coagulation, ELISA

## Abstract

The intravenous administration of polyclonal antibodies known as antivenom is the only effective treatment for snakebite envenomed victims, but because of inter-specific variation in the toxic components of snake venoms, these therapies have variable efficacies against different snake species and/or different populations of the same species. In this study, we sought to characterize the in vitro venom binding capability and in vitro cross-neutralizing activity of antivenom, specifically the Hemato Polyvalent antivenom (HPAV; The Queen Saovabha Memorial Institute (QSMI) of the Thai Red Cross Society, Thailand) and three monovalent antivenoms (QSMI) specific to *Daboia siamensis*, *Calloselasma rhodostoma,* and *Trimeresurus albolabris* venoms, against a variety of South Asian and Southeast Asian viper venoms (*Calloselasma rhodostoma*, *Daboia russelii*, *Hypnale hypnale*, *Trimeresurus albolabris*, *Trimeresurus purpureomaculatus*, *Trimeresurus hageni,* and *Trimeresurus fucatus*). Using ELISA and immunoblotting approaches, we find that the majority of protein components in the viper venoms were recognized and bound by the HPAV polyvalent antivenom, while the monospecific antivenom made against *T.*
*albolabris* extensively recognized toxins present in the venom of related species, *T. purpureomaculatus*, *T. hageni,* and *T. fucatus*. In vitro coagulation assays using bovine plasma revealed similar findings, with HPAV antivenom significantly inhibiting the coagulopathic activities of all tested viper venoms and *T. albolabris* antivenom inhibiting the venoms from Malaysian arboreal pit vipers. We also show that the monovalent *C. rhodostoma* antivenom exhibits highly comparable levels of immunological binding and in vitro venom neutralization to venom from both Thailand and Malaysia, despite previous reports of considerable intraspecific venom variation. Our findings suggest that Thai antivenoms from QSMI may by useful therapeutics for managing snake envenomings caused by a number of Southeast Asian viper species and populations for which no specific antivenom currently exists and thus should be explored further to assess their clinical utility in treating snakebite victims.

## 1. Introduction

Snakebite envenoming is an environmental, occupational, and climatic hazard that predominantly affects the rural, impoverished populations of low- and middle-income countries found in the tropics. The highest burden of snakebite exists in agricultural regions of Asia (i.e., South Asia and Southeast Asia), Papua New Guinea, sub-Saharan Africa, and Latin America [1]. Global estimates of the incidence of snakebite suggest that over 1,800,000 people are envenomed annually, resulting in between 90,000 and 138,000 deaths [2,3].

A number of snakes from the family Viperidae (“vipers”) are medically important in South Asia and Southeast Asia. There are two viper subfamilies, Viperinae (true vipers; e.g., Russell’s vipers) and Crotalinae (pit vipers; e.g., green pit viper, Malayan pit viper), with the latter named after their specialized sensory organ, the loreal pit, which detects infrared [4]. Human envenoming by both viper subfamilies may result in life-threatening outcomes, including disseminated intravascular coagulopathy, severe hemorrhage, and nephrotoxicity [4,5].

To combat the life-threatening and morbidity-causing effects of snakebite envenoming, two similar types of therapeutics, known as antivenom, are manufactured. Both consist of polyclonal antibodies sourced from hyper-immune animal plasma/sera, with monovalent antivenoms designed specifically for treating envenoming caused by a single specific snake species, and polyvalent antivenom covering a greater breadth of venom components found in snakes from more than one species. Typically, polyvalent antivenoms are more cost-effective to produce and provide a desirable option clinically, as they obviate the requirement to accurately identify the biting snake species [6]. However, due to reduced antibody specificity against any one snake venom, larger therapeutic doses of polyvalent antivenoms are often required to effect cure, which may increase the risk of adverse reactions and cost to patients [7].

Despite being more restricted in terms of their neutralizing breadth, monovalent antivenoms are the therapeutic of choice in regions where a single snake species is responsible for the majority of severe envenomings. Furthermore, several studies have demonstrated that monovalent products from one region can often exhibit cross-reactivity and neutralize venom toxicities from similar species found in other regions [8,9,10,11]. However, this is not always the case, as both inter- and intra-specific variation in venom toxin composition can result in a loss of antivenom efficacy or even therapeutic failure [12,13,14,15]. Despite the challenges associated with the composition of venom being unique to each snake species, many of the same medically important toxin families are found consistently represented within snake families. For example, coagulopathic toxins such as isoforms of the snake venom metalloproteinase (SVMP) and phospholipase A_2_ (PLA_2_) toxin families are typically abundant components of viper venoms [16], and though there is much variation in the toxin isoforms found within these venoms, structural similarities among such isoforms can result in cross-neutralization by antivenoms raised against different species [10,17,18].

The administration of antivenom is currently the only effective treatment for systemic envenoming (e.g., respiratory paralysis, hemorrhage, coagulopathy, etc.) [4]. Since there is no antivenom manufacturer in some Asian countries, most antivenom is imported from overseas, such as India and Thailand. In Southeast Asia, systemic snake envenoming is often treated with monovalent or polyvalent antivenoms manufactured by The Queen Saovabha Memorial Institute (QSMI) of the Thai Red Cross Society, Thailand. Indeed, the QSMI manufactures three monospecific antivenoms against three medically important vipers in Thailand; *D. siamensis* antivenom (DSAV), *C. rhodostoma* antivenom (CRAV), and *T. albolabris* antivenom (TAAV). Moreover, QSMI also manufacturers a polyvalent antivenom (Hemato Polyvalent Snake antivenom (HPAV)) against the venom of these same three snake species.

While the cross-neutralizing activity of these Thai antivenoms against the hemotoxicity of certain viper venoms has previously been explored in both in vitro and in vivo studies [5,9,19,20], their comparative cross-reactivity against a wide coverage of viper venoms has yet to be fully investigated. Specifically, it remains unclear the extent to which each of these commonly used snakebite therapies is capable of binding to the varying toxin components found across both homologous and heterologous snake species found in the region. In addition, it is worth noting that the capability of antivenom to recognize venom toxin components does not necessarily indicate that the antivenom is capable of neutralizing toxin-induced effects [10,21].

Consequently, the objective of this study was to determine the in vitro immunological binding and the in vitro coagulopathy neutralization of these four antivenoms raised against hemotoxic viper venoms from Thailand to cross-react with and inhibit venoms sourced from other viper species found in neighboring Asian countries (Malaysia and Sri Lanka). Our results demonstrate varying degrees of immunological cross-reactivity and venom neutralization against several Asian viper venoms, indicating that certain combinations of the QSMI antivenoms may potentially provide clinical benefit to snakebite victims in countries where there is a current absence of specific antivenom.

## 2. Results

### 2.1. Immunological Cross-Reactivity Determined by End-Point Titration (EPT) ELISA

We first assessed the ability of the various antivenoms (monovalent DSAV, CRAV, and TAAV, and polyvalent HPAV) to recognize and bind to the components present in a variety of pit viper venoms by ELISA (Table 1 and Figure 1). The snake species tested from Malaysia were *C. rhodostoma* (Malayan pit viper), *T. hageni* (Hagen’s pit viper), *T. purpureomaculatus* (Mangrove pit viper), and *T. fucatus* (Siamese Peninsular pit viper), and we used venom from Thai *C. rhodostoma* and *T. albolabris* (white-lipped green pit viper) as comparators.

The optical density (OD) readings (405 nm) of the antivenoms at the 1:2,500 dilution, in the middle of the downward slope displayed in the full binding profile shown in Figure 1, provide the most immunologically meaningful comparison and, for clarity, are displayed in Table 1. These data indicated that all antivenoms exhibited marked cross-reactivity to each of the venoms, although concentration-dependent decreases in binding varied between the antivenoms and the different venoms used. Overall, the polyvalent HPAV antivenom showed high levels of binding to each of the pit viper venoms tested. This result is perhaps not unexpected, given that this antivenom has the greatest breadth of immunogens used to raise the antibodies (i.e., multiple venoms), and the high levels of binding against *T. hageni*, *T. purpureomaculatus*, and *T. fucatus* suggests extensive paraspecific recognition of their venom toxins, since none are used as venom immunogens. Conversely, the DSAV antivenom showed low levels of binding to each of these venoms, suggesting that the venom composition of *D. siamensis*, which is used to make this therapeutic, is distinct from that of the Malaysian venoms tested here. For *C. rhodostoma*, extensive and comparable cross-reactivity was observed between the CRAV antivenom with the venoms from both Thailand and Malaysia. For the remaining species (*T. hageni*, *T. purporeomaculatus,* and *T. fucatus*), high binding levels were observed with the monospecific TAAV antivenom, and these were comparable to those obtained with the venom used during antivenom production (*T. albolabris*) and also the results obtained with use of the HPAV antivenom (Table 1 and Figure 1).

The same immunological comparisons using venoms from the two Sri Lankan true vipers, *D. russelii* and *H. hypnale*, also revealed high cross-reactivity with the HPAV polyvalent antivenom for both venoms and at levels comparable with the other species tested here. The DSAV monovalent antivenom also exhibited extensive recognition of *D. russelii* venom, which is perhaps unsurprising given that it is made using the venom of the congener *D. siamensis*, but it cross-reacted poorly with venom from *H. hypnale*. The CRAV antivenom, followed by the TAAV antivenom, showed increased cross-reactivity against *H. hypnale* with the former approaching the levels observed with the polyvalent HPAV product. In combination, these data indicate that the antibodies present in the varying Thai monospecific antivenoms are capable of substantial recognition and cross-reactivity with the heterologous venoms of other south and southeast Asian pit vipers, and that the polyvalent HPAV antivenom exhibits high levels of immunological binding against all species tested.

### 2.2. Immunological Avidity Determined by Chaotropic ELISA

We used a relative avidity ELISA to determine the strength of venom–antivenom binding interactions, specifically by incubating venom and antivenom with the chaotropic agent ammonium thiocyanate (NH_4_SCN), which is a potent disruptor of protein–protein interactions. The standardized antivenom solutions (1:1000 dilutions of 50 mg/mL concentrations) were incubated with the various viper venoms at the same concentration as for EPT ELISA (10 µg venom in 10 mL coating buffer), and the OD was quantified following the addition of increasing concentrations of 0, 1, 2, 3, 4, 6, and 8 M of NH_4_SCN.

The most immunologically informative results were obtained by the comparison of the percentage reduction in OD values after incubation with 4M NH_4_SCN (Table 2 and Figure 2). Our data indicate that HPAV displayed the highest, cross-snake species antivenom–venom binding avidities of all the antivenoms, as evidenced by the lowest percentage reductions in binding in the presence of the chaotrope (12.2–34.9% reductions; Table 2). The DSAV antivenom exhibited high avidity against Sri Lankan *D. russelii* venom, with only an 8.2% reduction observed, thus outperforming the polyvalent comparator (15.2% reduction). However, the DSAV monovalent performed poorly against all other venoms tested, with percentage reductions of >60% observed, with the exception of *H. hypnale* (47.2% reduction) (Table 2). Consistent with the results of the EPT ELISA, the TAAV antivenom exhibited highly comparable binding avidities to the HPAV for all of the arboreal pit vipers tested, including *T. fucatus* (19.5–31.8% reductions), but it performed less well against venoms from *C. rhodostoma*, *D. russelii,* and *H. hypnale* (39.0–65.1% reductions) (Table 2). As anticipated, the CRAV antivenom exhibited the highest avidities against the venoms of *C. rhodostoma*, although to our surprise, we observed stronger avidity toward venom from the Malaysian rather than the Thai locale used to generate the antivenom (12.9% vs. 27.0% reduction, respectively) (Table 2). The CRAV antivenom also displayed moderate binding avidities with venom from *H. hypnale* and *T. albolabris* (30.4% and 33.3% reduction, respectively), although the strength of binding interactions observed with the remaining arboreal pit viper venoms and that of *D. russelii* were considerably lower (all > 55% reduction) (Table 2).

### 2.3. Immunological Cross-Reactivity Visualized by Immunoblotting: South-East Asian Pit Viper Venoms

Visualization of venom protein–antibody binding was performed using immunoblotting with the venoms of the six southeast Asian pit vipers (Thai *C. rhodostoma*, Malay *C. rhodostoma*, *T. albolabris, T. hageni, T. purpureomaculatus*, and *T. fucatus*) and the four antivenoms (HPAV, DSAV, CRAV, and TAAV). First, the six venoms were separated by SDS-PAGE gel electrophoresis to characterize the toxin profiles (Figure 3A). The venoms displayed a wide range of molecular weight proteins, from ≈10 kDa to ≈100 kDa in mass. High-intensity bands, indicative of abundant venom proteins, were particularly noticeable in all venoms in the ≈10–35 kDa molecular weight range.

Western immunoblotting of the venom protein gels with the HPAV antivenom revealed extensive immunological cross-reactivity between the antivenom antibodies and the majority of the separated venom proteins, including both geographical variants of *C. rhodostoma* and the venoms of *T. albolabris* and *T. fucatus* (Figure 3B). However, despite the earlier findings from ELISA experiments, we note that both the diversity of toxin cross-reactivity and intensity of binding observed with the HPAV antivenom was much lower for venoms from *T. purpureomaculatus* and *T. hageni* (Figure 3B). Immunoblotting with CRAV revealed a highly equivalent recognition of the venom proteins found in the two *C. rhodostoma* venoms but a near complete absence of binding to the toxins found in the venoms of the other snakes tested (Figure 3C). In line with the low levels of binding observed in the ELISA experiments, immunoblotting with DSAV revealed very little immunological recognition against all of the southeast Asian pit viper venoms tested (Figure 3D). Finally, the monospecific *T. albolabris* antivenom (TAAV) recognized the majority of the venom proteins found in the arboreal pit viper species tested, particularly those of *T. albolabris, T. purpureomaculatus,* and *T. fucatus*, although lower toxin recognition was observed with venom from *T. hageni*, and only a few protein bands were visualized in the two *C. rhodostoma* venoms (Figure 3E).

### 2.4. Immunological Cross-Reactivity Visualized by Immunoblotting: Sri Lankan Viper Venoms

Following the same approach outlined above, *D. russelii and H. hypnale* venoms from Sri Lanka were resolved in an SDS-PAGE gel under reducing conditions (Figure 3F). Subsequent analysis of these venoms showed considerable variation in both the molecular weight and intensity of the resulting protein bands (Figure 3F), although the broad molecular weight range was comparable with that of the southeast Asian vipers. Noticeably, both venoms contained highly abundant proteins of <15 kDa in size, although the molecular weights differed by species.

In line with the EPT and avidity ELISA results, Western blotting analysis showed that HPAV was able to detect the vast majority of the proteins observed in both venoms, and in the case of *D. russelii*, it revealed additional venom proteins not detectable by SDS-PAGE (Figure 3G). Unsurprisingly, DSAV also displayed extensive immunological recognition to Sri Lankan *D. russelii* venom, although little recognition of *H. hypnale* venom was observed (Figure 3H). Conversely, CRAV extensively recognized the diversity of proteins found in *H. hypnale* venom but only a low number of specific bands in *D. russelii* venom (Figure 3I). The TAAV monovalent antivenom recognized a number of different protein components found in both of the Sri Lankan viper venoms but exhibited lower binding intensities than HPAV and the DSAV monovalent vs. *D. russelii* and CRAV monovalent vs. *H. hypnale* (Figure 3J).

### 2.5. Coagulopathic Venom Activity and Antivenom Neutralization Measured by Plasma Clotting Assay

The venoms (500 ng) of *C. rhodostoma* (Thailand and Malaysia), *T. albolabris, T. hageni, T. purpureomaculatus, D. russelii*, and *H. hypnale* all caused rapid coagulation of 20 µL bovine plasma within 20 min when compared with the phosphate-buffered saline (PBS) control (Figure 4). Interestingly, the venom of *T. fucatus* at the same concentration (500 ng) delayed plasma coagulation activity, suggesting that this venom is anticoagulant, rather than procoagulant (Figure 4). Next, we sought to assess whether the various antivenoms were capable of neutralizing these coagulopathic venom activities. First, we demonstrated that none of the antivenoms (i.e., HPAV, CRAV, DSAV, and TAAV) used in this study had a major effect on coagulation at the doses tested in the absence of venom (Appendix A). Next, we tested the ability of the antivenoms to neutralize the coagulopathic activity of the two *C. rhodostoma* venoms. Both the HPAV and CRAV, tested at 1× titer recommended by the manufacturer (1 mL of antivenom neutralizing 1.6 mg venom), were found to significantly inhibit the rapid procoagulant activity of *C. rhodostoma* venom to control (PBS) levels, irrespective of the Thai or Malaysian origin of the venom (Figure 4A,B). Contrastingly, neither the DSAV or TAAV antivenoms showed inhibitory activities against the two *C. rhodostoma* venoms, which is a finding that is highly consistent with the low levels of binding observed in the various immunological assays (Figure 4A,B).

In an analogous manner, the procoagulant activity of *T. albolabris, T. hageni*, and *T. purpureomaculatus* venoms were significantly inhibited by both the polyvalent HPAV and the monovalent TAAV antivenom when used at doses scaled to 1× their recommended titer (1 mL of antivenom neutralizing 0.7 mg *T. albolabris* venom) (Figure 4C–E). Surprisingly, the DSAV and CRAV antivenoms were also capable of significantly reducing the coagulopathic activity of *T. albolabris* and *T. purporeomaculatus* venoms, although not to the same degree as the HPAV and TAAV antivenoms (Figure 4C,E). The CRAV antivenom was noticeably less effective at neutralizing the coagulopathic venom activity of *T. hageni*, although the DSAV antivenom surprisingly outperformed that of the TAAV product (Figure 4D). Despite the venom of *T. fucatus* being anticoagulant rather than procoagulant, both the HPAV and TAAV antivenoms were also found to significantly inhibit this activity, and they restored coagulation times to levels highly comparable with the control (Figure 4F). However, neither DSAV or CRAV antivenoms exhibited any inhibitory activity against the coagulotoxicity of *T. fucatus* venom.

Both of the Sri Lankan viper venoms tested (*D. russelii* and *H. hypnale*) exhibited procoagulant venom effects, although the neutralization of coagulotoxicity varied among the antivenoms used (Figure 4G,H). Both the HPAV and DSAV antivenoms completely inhibited the procoagulant effects of *D. russelii* venom, but they resulted in a net anticoagulant effect (i.e., delayed clotting compared with the PBS control), suggesting that some anticoagulant venom toxins may not be neutralized by these therapeutics (Figure 4G). Neither the CRAV or the TAAV antivenom exhibited any inhibitory effect against the procoagulant effect of *D. russelii* venom. At the manufacturer’s recommended therapeutic dose for *C. rhodostoma* venom, HPAV significantly decreased the procoagulant effect of *H. hypnale* venom, although not to control levels (Figure 4H). Surprisingly, no significant neutralization of *H. hypnale* venom was observed with CRAV in this study, despite promising levels of immunological recognition, while neither of the other two monovalent antivenoms showed any inhibitory activity against this venom (Figure 4H)

## 3. Discussion

Snakebite envenoming is an occupational hazard in many tropical countries and is responsible for causing a variety of life-threatening effects including neurotoxicity, myotoxicity, systemic coagulopathy, hemorrhage, and renal failure. The mainstay of snakebite therapy is the administration of animal-derived antivenom. Unfortunately, the high production cost of antivenom, coupled with weak demand in impoverished countries, has led to a number of manufacturers discontinuing the manufacture of snakebite therapies, resulting in a shortage of effective treatment in many areas of the world. One of the additional challenges associated with snakebite treatment is ensuring the correct identification of the snake species responsible for a bite, and thus making an informed decision relating to the appropriate antivenom therapy for use (e.g., species-specific monovalent vs. polyvalent). This is important because the administration of large volumes of non-specific antivenom not only can result in treatment failure but also increases the risk of antivenom adverse reactions, which include severe hypersensitivity and serum sickness [22,23]. Consequently, polyvalent antivenoms are frequently used to circumvent issues associated with incorrect antivenom use, despite typically requiring higher therapeutic doses to effect cure [7]. In Southeast Asia, previous studies have demonstrated that the Thai Hemato Polyvalent antivenom (HPAV) and corresponding monovalent antivenoms were capable of neutralizing the in vivo lethal effects and certain in vitro toxicities of a small number of medically important snakes [9,19]. In the present study, we expanded the scope of these initial studies to investigate, in a standardized manner, the cross-reactivity of homologous and heterologous antivenoms from Thailand to a variety of south and southeast Asian viper venoms using immunological-binding and in vitro neutralization methods. Our data reveal the titer, avidity, and specificity of immunological binding of each antivenom to heterologous and homologous venoms from Malaysia and Sri Lanka, and they suggest that the HPAV antivenom should be explored further as a potentially useful therapeutic for treating snakebite across the region.

In Malaysia, over 350 snakebite cases were admitted to hospitals in Kuala Lumpur [24] and Western Malaysia [25] between 1999 and 2003, although none of the treated bites resulted in mortality. These positive outcomes seem likely to be the result of good medical access, appropriate health-seeking behaviors, and the accessibility of snakebite therapeutics in these areas. However, in other regions of Malaysia, and throughout much of Southeast Asia, fatalities are more common. For example, severe envenoming, resulting in impaired respiration as the result of neuromuscular paralysis or intracerebral hemorrhage [26], has been reported following snake envenoming in East Malaysia (Borneo) [27] and a number of Indonesian islands [28], where access to medical resources and snakebite therapies are limited. Cobras (*Naja* spp.) and vipers are recognized as the most medically important snakes in terms of causing morbidity and mortality [25], with the latter group containing the Malayan pit viper (*C. rhodostoma*), white-lipped green pit viper (*T. albolabris*), and Russell’s viper (*D. siamensis*), which are categorized as the most clinically significant vipers of Southeast Asia [4]. However, the presence of Russell’s viper has not been reported in Peninsular Malaysia nor Borneo [29]. Additionally, envenoming by less studied “green pit vipers” such as the mangrove pit viper (*T. purpureomaculatus*), Hagen’s pit viper (*T. hageni*), and the Siamese Peninsular pit viper (*T. fucatus*: former known as *Popeia fucata*) have been shown to cause severe bleeding disorders and tissue necrosis, resulting in amputation [30]. Since there is no antivenom manufacturer in Malaysia, most antivenom is currently imported from Thailand, yet a comprehensive understanding of the interaction of these antivenoms with Malaysian snake venoms is lacking.

In the present study, we show that the HPAV antivenom extensively recognizes and cross-reacts with a diversity of toxins present in pit viper venoms from Malaysia, while the CRAV monovalent cross-reacts well with Malaysian *C. rhodostoma* venom, and the TAAV cross-reacts with venom from various Malaysian arboreal pit viper species. Our ELISA experiments demonstrated that HPAV binds strongly to all of the tested venoms, with the exception of *T. fucatus* venom, where moderate binding was observed. High concentrations of the various monospecific antivenoms (i.e., DSAV, CRAV, and TAAV) were required before comparable immunological cross-reactivity with heterologous venoms was observed, and these antivenoms displayed a much more limited breadth of cross-reactivity, as anticipated (e.g., CRAV vs. Malaysian *C. rhodostoma*, TAAV vs. the Malaysian arboreal pit vipers). The data generated by the avidity ELISA using the same antivenom and venom combinations were largely consistent with the results of the end-point titration ELISA, with the HPAV antivenom exhibiting impressively low reductions in antibody binding in the presence of the chaotrope, and with reductions in binding greatest (i.e., lowest avidity) against the venom of *T. fucatus*. Surprisingly, with the monovalent antivenoms, we observed some cases where the strength of antibody binding against heterologous venoms superseded that of the venom used for immunization (e.g., CRAV vs. Malaysian *C. rhodostoma*, TAAV vs. *T. purporeomaculatus*). A recent study showed a high level of similarity between the venom proteomes of Malaysian *T. purpureomaculatus* and Thai *T. albolabris*, which may underpin the basis of this latter interaction [31].

While high levels of immunological binding are an essential prerequisite for an effective antivenom, binding does not always result in venom neutralization [10,21]. To investigate whether the high levels of binding observed between the HPAV polyvalent antivenom, and to a lesser extent the monovalent antivenoms, and Malaysian snake venoms result in any form of venom neutralization, we used an in vitro venom coagulation assay. This is relevant because clinical outcomes observed following envenoming by Asian pit vipers include systemic coagulopathy and hemorrhage, as well as progressive edema and local tissue necrosis, which can result in amputation of the affected digit or limb [22,30,32,33]. It has been postulated that venom thrombin-like serine protease (SVSP) and snake venom metalloproteinase (SVMP) toxins play an important role in causing the coagulopathy observed following envenomings by pit vipers [33,34]. Our coagulation assay clearly demonstrated that the venoms of *C. rhodostoma* and the *Trimeresurus* spp. (with the exception of *T. fucatus*) cause rapid coagulation of bovine plasma, and that these venom activities were all significantly inhibited (to baseline levels) by the HPAV polyvalent antivenom. In addition, the monovalent CRAV neutralized the coagulopathic venom activity of both geographical variants of *C. rhodostoma*, while the TAAV product significantly reduced the coagulotoxicity of the *Trimeresurus* species. This latter observation correlates with recent comprehensive observations of the potential utility of the TAAV monovalent antivenom against coagulopathy caused by certain *Trimeresurus* spp. [20], although here we found a degree of neutralization against *T. hageni* venom (although less than observed with the other *Trimeresurus* spp. tested) that was not apparent from those previous findings. Despite seemingly low levels of immunological cross-reactivity, the DSAV monovalent antivenom also significantly reduced procoagulation caused by the various *Trimeresurus* venoms. This result is surprising given that our immunoblotting experiments revealed only low levels of toxin recognition by this antivenom with these venoms, including only one band >20 kDa in size that might correspond with an SVSP or SVMP toxin. Further research is required to understand the basis of this surprising result. The final Malaysian venom tested, that of *T. fucatus*, exhibited anticoagulant, rather than procoagulant, activity. Consistent with our immunological studies, both HPAV and TAAV antivenoms restored venom-induced coagulation to control levels, while neither the CRAV nor DSAV monovalent products had a significant effect. These findings suggest that *T. fucatus* venom contains anticoagulant proteins that exhibit similar antigenic properties to components present in the venom of *T. albolabris*, although venom compositional profiling (e.g., via transcriptomics and proteomics) is required to robustly test this hypothesis.

In Sri Lanka, the Russell’s viper (*D. russelii*) and hump-nosed pit viper (*H. hypnale*) are classified as category 1 medically important venomous species as a result of their capability to cause severe systemic envenoming [4]. Unfortunately, commercial snake antivenom manufactured using native Sri Lankan snake venoms as the immunogens is not currently available, although a new polyspecific product is in development and awaiting clinical evaluation of its safety and effectiveness [35]. Consequently, in the meantime, polyvalent antivenoms produced by Indian manufacturers against Indian snake venoms have been imported for treating envenoming in the country. However, the venom of *H. hypnale* is not included in the immunization mixture of this polyvalent antivenom. Moreover, these Indian venoms likely contain at least some degree of geographical variation in venom composition, which might negatively impact upon treatment efficacy [15]. In this study, Sri Lankan *D. russelii* and *H. hypnale* venoms displayed high levels of immunological binding, in terms of both titers and avidities, to the DSAV and CRAV monovalent antivenoms, respectively. In addition, the HPAV product exhibited equivalent levels of antibody–protein binding and similar avidities to the monovalent antivenoms against both venoms, and they also significantly reduced their procoagulant venom effects, though not to control levels for either venom. Contrastingly, the CRAV did not cause significant inhibition of *H. hypnale* venom-induced coagulation, despite high levels of immunological cross-reactivity, and this antivenom previously being demonstrated to effectively neutralize a variety of toxicities caused by Sri Lankan *H. hypnale* venom, including murine in vivo lethality and procoagulation [36]. This discrepancy is likely the result of an insufficient dose of antivenom being used in our assay to observe an inhibitory effect, and thus while this antivenom may still prove to be a useful treatment for *H. hypnale* envenomings [36] in the absence of any specific antivenom, higher therapeutic doses than those recommended for treating *C. rhodosomta* envenomings are likely to be required, but they first require robust clinical testing. Our results also demonstrated that the DSAV monovalent antivenom made against Thai *D. siamensis* venom exhibits high levels of immunorecognition against the venom of the sister species *D. russelii* sourced from Sri Lanka. Moreover, this antivenom also significantly inhibited the procoagulant venom effects of *D. russelii* venom, although this inhibition resulted in a net anticoagulant venom effect (also observed with HPAV), suggesting that anticoagulant toxins may not be being neutralized. Nonetheless, these findings suggest that Sri Lankan *D. russelii* venom shares a number of similar antigens with *D. siamensis* from Thailand, as previously suggested [36]. These results also extend the findings of our previous study, which demonstrated that both HPAV and DSAV displayed potentially wide geographical utility against a variety of *D. siamensis* venoms sourced from distinct locales and therefore may be useful tools for managing snakebite envenomings by Russell’s vipers where there is an absence of locally manufactured antivenoms [5].

In summary, our in vitro immunological experiments show that the Hemato polyvalent antivenom (HPAV) from Thailand exhibits substantial cross-reactivity with the venoms of a variety of Malaysian pit vipers, particularly snakes of the genera *Calloselasma* and *Trimeresurus*. Monospecific antivenom raised against Thai *T. albolabris* venom (TAAV) also displayed high immunological cross-reactivity to all of the arboreal pit viper venoms tested, providing evidence consistent with the current Malaysian guideline on the management of snakebite for supporting the use of TAAV in envenomed victims where species-specific antivenom is unavailable. The HPAV antivenom also displayed immunological cross-reactivity and a degree of venom neutralization against Sri Lankan viper venoms and thus may be of potential clinical benefit in the current absence of Sri Lankan-specific antivenom. However, additional experimental studies, specifically in vivo preclinical efficacy experiments followed by clinical assessments of antivenom safety and efficacy, are required prior to routine human use of heterologous antivenoms for treating South Asian and Southeast Asian snakebite victims.

## 4. Materials and Methods

### 4.1. Snake Venom and Antivenoms

Venoms of Malaysian pit vipers (*C. rhodostoma*, *T. purpureomaculatus*, *T. hageni*, and *T. fucatus*) were milked from specimens captured in Northwest Peninsular Malaysia. The venom from three individuals from each species was extracted by allowing the snakes to bite plastic containers wrapped with parafilm. The specimens were milked three times with a time interval of three weeks between milking before being released at the area of capture. Then, pools of venom from each species were generated, frozen, and freeze-dried. The research permit for Malaysian snakes was provided by the Department of Wildlife and National Parks, Government of Malaysia (permit number: HQ-00067-15-70). For the Thai pit vipers (i.e., *C. rhodostoma* and *T. albolabris*) and Sri Lankan vipers (*D. russelii* and *H. hypnale*), we used lyophilized venoms stored in the historical venom collection of the Centre for Snakebite Research and Interventions at the Liverpool School of Tropical Medicine. Due to the age of the samples (>20 years old), no knowledge of the number of specimens contributing to each species pool used was known. Freeze-dried venom samples were weighed, labeled, and stored at −20 °C prior use. When required, venoms were weighed, reconstituted in phosphate-buffered saline (PBS) and venom protein concentrations measured using a Nanodrop (ThermoFisher, Fitchburg, WI, USA) and BCA protein assay (Pierce Biotechnology, Rockford, IL, USA).

Monovalent *D. siamensis* antivenom (DSAV; Lot No.: WR00117), *C. rhodostoma* antivenom (CRAV; Lot No.: CR00316), and *T. albolabris* antivenom (TAAV; Lot No.: TA00317), as well as the Hemato Polyvalent antivenom (HPAV; Lot No.: HP00218), were purchased from QSMI. All antivenoms are equine F(Ab’)_2_-based products. Freeze-dried antivenoms were dissolved with pharmaceutical grade water supplied by the manufacturer according to their instructions. The dissolved antivenoms were stored at 4 °C prior to use. The concentration of the reconstituted antivenoms, as measured by a Nanodrop (ThermoFisher, Fitchburg, WI, USA), were HPAV, 54 mg/mL; DSAV, 21 mg/mL; CRAV, 24.5 mg/mL; and TAAV, 14 mg/mL.

### 4.2. Immunological Assays

All immunological assays were performed as previously described [11,21]. The protein concentrations of antivenom were standardized to 50 mg/mL for all immunological assays to permit direct comparisons.

#### 4.2.1. End-Point Titration (EPT) ELISA

ELISA plates (96 wells) were coated with 100 ng of venom (a separate plate for each snake species) in carbonate buffer (sodium bicarbonate buffer 0.075 M, with sodium carbonate buffer 0.025 M at room temperature), pH 9.6, and then incubated at 4 °C overnight. Plates were washed using six changes of Tris-buffered saline buffer (TBST) (0.01 M Tris-HCl, pH 8.5; 0.15M NaCl; 1% Tween 20) and incubated at room temperature for three hours with 5% non-fat milk (diluted with TBST) to “block” non-specific reactivity. Then, the plates were washed and incubated (in duplicate) with 100 µl monospecific antivenoms or HPAV at an initial dilution of 1:100 followed by 1:5 serial dilutions (1:100 to 1:39,063,500 dilutions) and incubated overnight at 4 °C. Commercially sourced IgG from non-immunized horses (Sigma-Aldrich, Gillingham, UK) was used at the same concentration as the negative control. The plates were washed again as described above and then incubated with 100 µl horseradish peroxidase-conjugated rabbit anti-horse IgG (1:1000; Sigma-Aldrich, Gillingham, UK) for two hours at room temperature. Following another TBST washing step, venom–antibody interactions were visualized by the addition of substrate (0.2% 2, 2′-Azino-bis (3-ethylbenzothiazoline-6-sulfonic acid) diammonium salt) in citrate buffer, pH 4.0, containing 0.015% hydrogen peroxide (Sigma-Aldrich, Gillingham, UK)) and incubated for 15 min prior to quantification. The optical density of the reaction was measured at 405 nm using an ELISA plate reader (LT-4500 automatic microplate absorbance reader). The most immunologically meaningful comparisons were displayed using the OD readings of the antivenoms at the 1:2500 dilution, in the middle of the downward slope.

#### 4.2.2. Relative Avidity ELISA

This assay was performed as described above for the end-point titration ELISA, except that each antivenom was diluted to a single concentration of 1:10,000 and then incubated overnight at 4 °C. Then, the incubated plates were washed with six changes of TBST followed by the addition of a chaotrope (ammonium thiocyanate, NH_4_SCN) in a range of concentrations (0–8 M) for 15 min. Next, the plates were washed again with TBST, and all subsequent steps were the same as the end-point titration ELISA. The relative avidity for venom–antivenom binding was determined as the percentage of reduction in ELISA OD reading (measured at 405 nm) between the maximum (8 M) and minimum (0 M) concentration of NH_4_SCN.

#### 4.2.3. SDS-PAGE

Each lyophilized venom was reconstituted to 1 mg/mL in reduced protein loading buffer (containing β-mercaptoethanol) and heated at 98.9 °C for 5 min. Venoms (10 µg) and molecular weight marker (Broad range molecular weight protein marker, Promega) were loaded onto 15% SDS-PAGE gels and then fractionated under 200 volts. The gel was stained using Coomassie Blue R-250 and visualized using a ChemiDoc XRS Imaging System (BioRad, Hercules, CA, USA).

#### 4.2.4. Western Blotting

The venoms were separated by electrophoresis as described above, except that after separation, the gels were transferred onto 0.45 µm nitrocellulose membranes by semi-dry electroblotting using a Bio-Rad Trans Blot Turbo (13 A, 25 V, 7 min) and as described in the manufacturer’s protocols (Bio-Rad, Watford, UK). Thereafter, the membranes were incubated overnight in blocking buffer (5% non-fat milk in TBST buffer), followed by six washes with TBST over 30 min and incubation overnight with primary antibody (monospecific antivenoms and HPAV) diluted 1:5000 in blocking buffer. Blots were washed again and then incubated for two hours with horseradish peroxidase-conjugated rabbit anti-horse secondary antibody (Sigma-Aldrich, Gillingham, UK) diluted 1:1500 in TBST. Following a final washing step with TBST, venom–antibody binding was visualized by the addition of DAB substrate (50 mg 3,3-diaminobenzidine, 100 mL PBS and 0.024% hydrogen peroxide; Sigma-Aldrich, Gillingham UK).

### 4.3. Plasma Coagulation Assay

Quantification of coagulopathic venom activity was performed in 384-well plate format, as previously described [37]. Frozen citrated bovine plasma (VWR International, Leicestershire, UK) was warmed in a water bath at 37 °C and centrifuged for 4 min at relative centrifugal force of 448 (2000 rpm) with an Eppendorf 5810 R centrifuge. Phosphate-buffered saline (PBS) (10 µL/well) was used as control (PBS alone) as well as a diluent. Stock solutions of venom (500 ng/10 µL) were manually added to separate wells in a 384-well microtiter plate. A fresh solution of 20 mM CaCl_2_ (20 µL/well) and the resulting platelet-poor plasma (20 µL/well) were then robotically pipetted to each well using a Thermo Scientific Multidrop 384-minirobot. To determine the protective activity of antivenom on coagulation, either HPAV (0.17 µL, 9.2 µg/well), DSAV (0.17 µL, 3.6 µg/well), CRAV (0.07 µL, 1.7 µg/well), or TAAV (0.15 µL, 3.6 µg/well) was added to the venom solution for 10 min prior to the addition of CaCl_2_ and plasma.

The prepared plate was then placed in a plate reader immediately following mixture via pipetting. Kinetic absorbance was measured at 25 °C every 76 s for 100 cycles at 595 nm using a BMG Fluostar Omega plate reader (BMG Labtech, Aylesbury, UK). Different sources of data, consisting of single reading and average rate in time per well, were obtained for the determination of coagulation curves. The area under the curve (AUC) of each reaction was calculated and normalized as the percentage of venom clotting activity.

### 4.4. Analysis of Results and Statistics

Statistical analyses of the resulting AUC data from the venom clotting experiments were performed using GraphPad Prism 6 (GraphPad Software Inc., San Diego, CA, USA). Multiple comparisons between venom, antivenom, and control conditions were performed using one-way analysis of variance (ANOVA) followed by Bonferroni’s multiple comparison test. Statistical significance was indicated where *p* < 0.05.

## Figures and Tables

**Figure 1 toxins-12-00766-f001:**
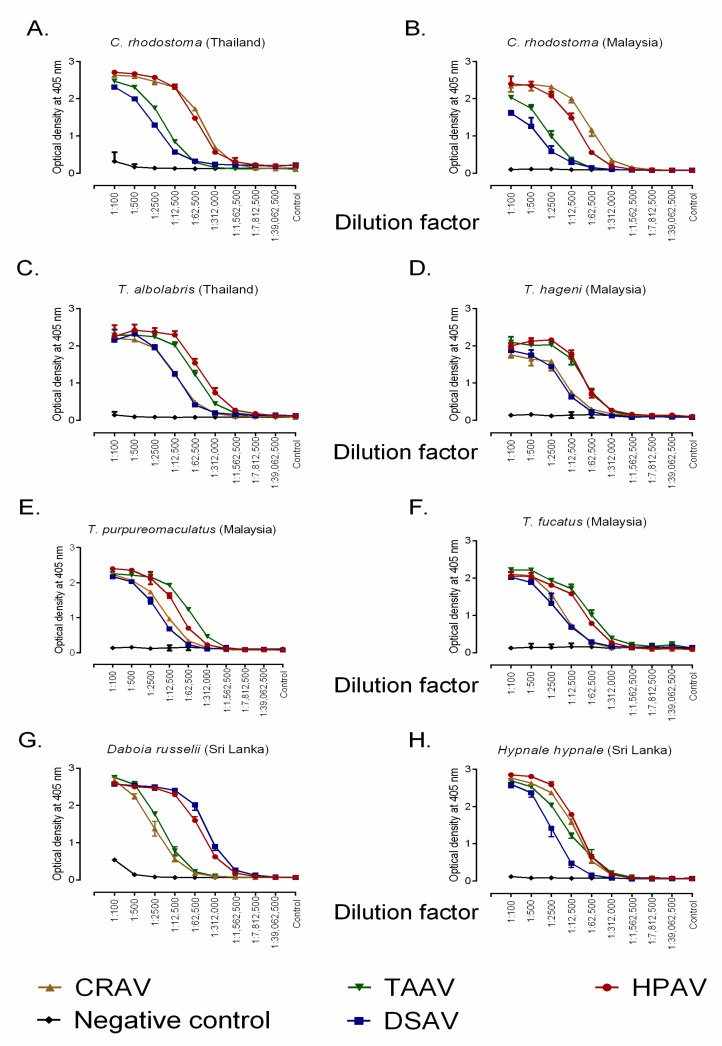
Quantification of the immunological binding between four commercial antivenoms manufactured by the Thai Red Cross Society and various south and southeast Asian snake venoms determined by end-point titration ELISA. Binding profiles for the four antivenoms, in addition to the negative control (horse IgG), are displayed against (**A**) *C. rhodostoma* venom from Thailand, (**B**) *C. rhodostoma* venom from Malaysia, (**C**) *T. albolabris* venom from Thailand, (**D**) *T. hageni* venom from Malaysia, (**E**) *T. purpureomaculatus* venom from Malaysia, (**F**) *T. fucatus* venom from Malaysia, (**G**) *D. russelii* from Sri Lanka, and (**H**) *H. hypnale* from Sri Lanka. All antivenoms were adjusted to 50 mg/mL in phosphate-buffered saline (PBS) prior to being diluted 1 in 100 and then serially diluted 1 in 5. Error bars indicate standard deviation of mean (SD). CRAV, *C. rhodostoma* antivenom; TAAV, *T. albolabris* antivenom; HPAV, Hemato Polyvalent antivenom; DSAV, *D. siamensis* antivenom.

**Figure 2 toxins-12-00766-f002:**
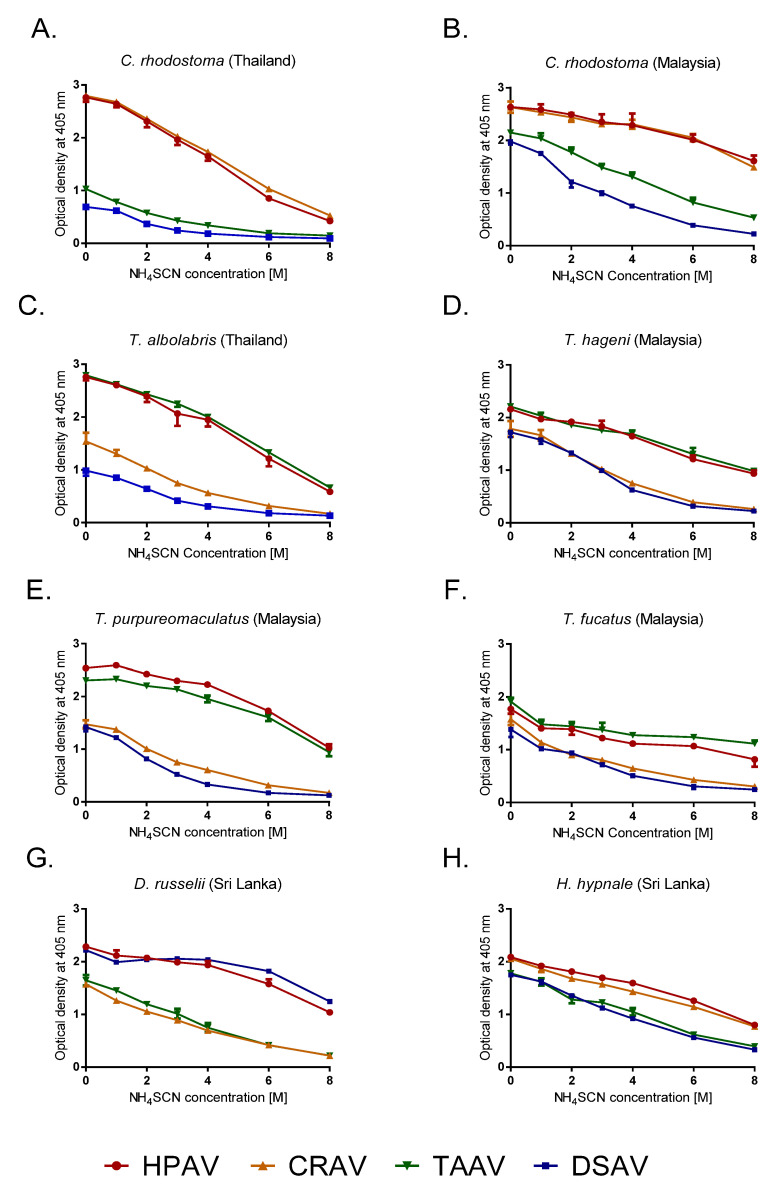
The relative avidity of four commercial antivenoms manufactured by the Thai Red Cross Society against various south and southeast Asian snake venoms determined by chaotropic ELISA. Avidity profiles for the four antivenoms incubated in the presence of increasing concentrations of the chaotrope (NH_4_SCN) are displayed for (**A**) *C. rhodostoma* venom from Thailand, (**B**) *C. rhodostoma* venom from Malaysia, (**C**) *T. albolabris* venom from Thailand, (**D**) *T. hageni* venom from Malaysia, (**E**) *T. purpureomaculatus* venom from Malaysia, (**F**) *T. fucatus* venom from Malaysia, (**G**) *D. russelii* from Sri Lanka, and (**H**) *H. hypnale* from Sri Lanka. All antivenoms were adjusted to 50 mg/mL in PBS and 1:1000 dilutions were used prior to incubation with increasing concentrations of NH_4_SCN 15 min. Error bars indicate standard deviation of mean (SD) as the triplicate readings. HPAV, Hemato Polyvalent antivenom; CRAV, *C. rhodostoma* antivenom; TAAV, *T. albolabris* antivenom; DSAV, *D. siamensis* antivenom.

**Figure 3 toxins-12-00766-f003:**
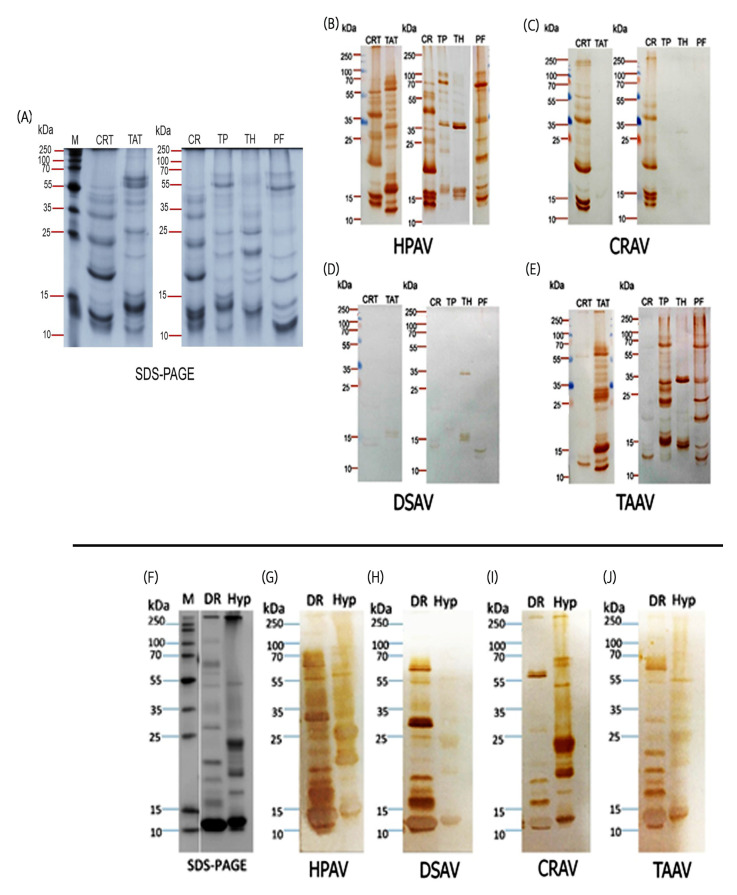
The protein profiles of the various Asian viper venoms used in this study and their immunological cross-reactivity with four antivenoms from the Thai Red Cross Society visualized by immunoblotting. (**A**) SDS-PAGE analysis of *C. rhodostoma* (CRT) and *T. albolabris* venoms (TAT) from Thailand (left panel) with protein marker (M), and *C. rhodostoma* (CR), *T. purpureomaculatus* (TP), *T. hageni* (TH) and *T. fucatus* (PF) venoms from Malaysia (right panel). (**B**–**E**) show the immunological cross-reactivity of the Hemato Polyvalent (HPAV) (**B**), *C. rhodostoma* monovalent (CRAV) (**C**), *D. siamensis* monovalent (DSAV) (**D**), and *T. albolabris* monovalent (TAAV) (**E**) antivenoms with those venoms by immunoblotting. (**F**) SDS-PAGE analysis of Sri Lankan *D. russelii* (DR) and *H. hypnale* (Hyp) venoms with protein marker (M). (**G**–**J**) Immunological cross-reactivity of the Hemato Polyvalent (HPAV) (**G**), *D. siamensis* monovalent (DSAV) (**H**), *C. rhodostoma* monovalent (CRAV) (**I**) and *T. albolabris* monovalent antivenoms (TAAV) (**J**) with those venoms by immunoblotting.

**Figure 4 toxins-12-00766-f004:**
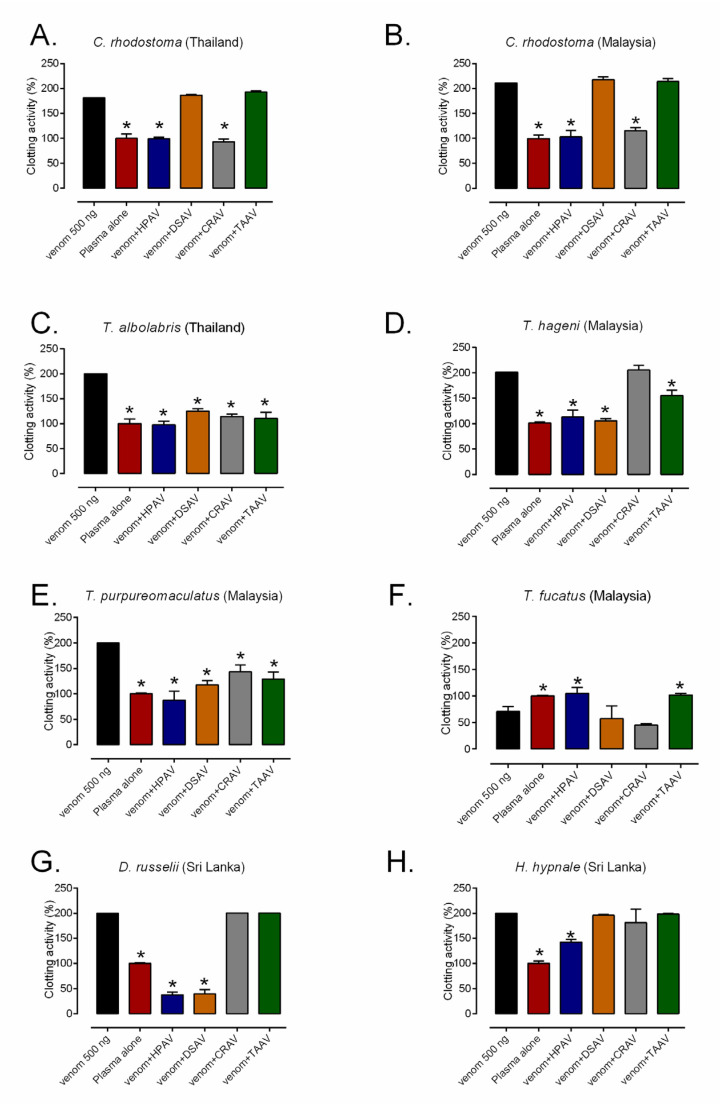
The procoagulant activity of the eight different viper venoms: (**A**) Thai *C. rhodostoma*, (**B**) Malaysian *C. rhodostoma*, (**C**) *T. albolabris*, (**D**) *T. hageni*, (**E**) *T. purpureomaculatus*, (**F**) *T. fucatus*, (**G**) *D. russelii,* and (**H**) *H. hypnale*, and their neutralization by the Hemato Polyvalent (HPAV), *D. siamensis* (DSAV), *C. rhodostoma* (CRAV), and *T. albolabris* (TAAV) antivenoms. The antivenoms were tested at the manufacturer’s recommended therapeutic dose. The data represent kinetic profiles of clotting from the plasma coagulation assay displayed as mean areas under the curve from triplicate measurements, which is transformed into percentage of the plasma control, and error bars on treatment groups represent SD. * *p* < 0.05, compared to venom alone (one-way ANOVA, followed by Bonferroni *t*-test).

**Table 1 toxins-12-00766-t001:** The optical density (OD) readings of antibody–venom protein interactions with various antivenoms at the discriminatory 1:2500 dilution as determined by end-point titration ELISA. ** indicates the venom from the same species and geographical origins as those used to raise the antibodies.

Venoms	Antivenom			
	Hematopolyvalent (HPAV)	*D. siamensis* (DSAV)	*C. rhodostoma* (CRAV)	*T. albolabris* (TAAV)
*C. rhodostoma* (Thailand) **	2.57 ± 0.03	1.29 ± 0.02	2.46 ± 0.08	1.75 ± 0.05
*C. rhodostoma* (Malaysia)	2.07 ± 0.11	0.58 ± 0.14	2.32 ± 0.18	0.98 ± 0.15
*T. albolabris* (Thailand) **	2.37 ± 0.10	1.95 ± 0.07	1.94 ± 0.02	2.24 ± 0.05
*T. hageni* (Malaysia)	2.16 ± 0.04	1.44 ± 0.06	1.65 ± 0.20	2.03 ± 0.06
*T. purpureomaculatus* (Malaysia)	2.13 ± 0.11	1.48 ± 0.08	1.73 ± 0.05	2.16 ± 0.07
*T. fucatus* (Malaysia)	1.81 ± 0.05	1.32 ± 0.26	1.57 ± 0.12	1.93 ± 0.01
*D. russelii* (Sri Lanka)	2.47 ± 0.04	2.50 ± 0.09	1.37 ± 0.19	1.77 ± 0.03
*H. hypnale* (Sri Lanka)	2.59 ± 0.08	1.42 ± 0.22	2.38 ± 0.02	2.03 ± 0.01

**Table 2 toxins-12-00766-t002:** Percentage reduction in OD after 4M NH_4_SCN treatment in the chaotropic ELISA assay. ** indicates the venoms used to raise the antibodies.

Venoms	Antivenom			
	Hematopolyvalent (HPAV)	*D. siamensis* (DSAV)	*C. rhodostoma* (CRAV)	*T. albolabris* (TAAV)
*C. rhodostoma* (Thailand) **	29.0 ± 4.0	75.0 ± 0.1	27.0 ± 1.1	65.1 ± 4.5
*C. rhodostoma* (Malaysia)	16.9 ± 4.0	61.9 ± 1.8	12.9 ± 5.5	39.0 ± 3.5
*T. albolabris* (Thailand) **	13.6 ± 1.0	60.2 ± 2.0	33.3 ± 5.1	19.5 ± 4.6
*T. hageni* (Malaysia)	23.7 ± 0.4	63.6 ± 2.6	57.8 ± 1.4	23.3 ± 1.6
*T. purpureomaculatus* (Malaysia)	12.2 ± 1.7	76.7 ± 1.0	58.7 ± 4.0	15.2 ± 1.5
*T. fucatus* (Malaysia)	34.9 ± 6.3	63.2 ± 3.2	59.0 ± 1.9	31.8 ± 1.2
*D. russelii* (Sri Lanka)	15.2 ± 4.9	8.2 ± 1.3	55.8 ± 1.9	54.5 ± 3.3
*H. hypnale* (Sri Lanka)	23.7 ± 0.8	47.2 ± 3.3	30.4 ± 0.4	40.9 ± 4.4

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
