# Peer review of "In Vitro Immunological Cross-Reactivity of Thai Polyvalent and Monovalent Antivenoms with Asian Viper Venoms"

_toxins, 2020, doi:10.3390/toxins12120766_

Round 1
Reviewer 1 Report
The manuscript describes the possible cross-reactivity of Thai antivenoms against Asian viper venoms, specifically Malaysian and Sri Lankan venoms, which are not included in the immunization mixture to produce the antivenoms tested here. Overall, I found the manuscript interesting since antivenom availability is a valid concern and crisis particularly in parts of Asia. However, I have some questions and want to address some comments that, in my opinion, should be taken into consideration to improve the quality of the manuscript.
Introduction:
In the sentence starting in line 83, is addressed the lack of information regarding the binding capability of the antivenoms towards the toxins found in venoms not used in the immunization mixture. Even though is really well explained in the discussion, it should be mentioned also in the introduction that the immune recognition of these toxins does not necessarily mean effective neutralization.
Results and discussion:
In the immunological cross-reactivity determination through EPT ELISA, how it was established the cut-off value of the assay to consider a binding as significant?
In the immunological cross-reactivity visualized by immunoblotting with the Southeast Asian pit viper venoms, the protein profiles from the SDS-PAGE gels in figure A differ a lot from the immunoblotting results. You should displayed the results using the same molecular weight markers for both figures, in order to facilitate the comparison.
Regarding the neutralization of the coagulant activity, I have some concerns:
a) You should add a control of antivenom only to assured that is not interacting with the plasma.
b) The plasma employed in the assays, was rich or poor of platelets? Based on this information, in the discussion you should mention or not the platelets.
c) Did you establish a minimal coagulation dose? I think this should be important since it could be the reason why you did not see neutralization of the activity displayed by H. hypnale, compared to the literature. Moreover, this would assure a better understanding of the neutralizing activity of the antivenoms tested here.
d) Why did you incubate the venoms with the antivenoms 10 min at room temperature and not as suggested by WHO when testing the efficacy of commercial antivenoms, which is 30 min at 37ºC?
e) In addition, the way the results are expressed can be confusing since 100% of coagulant activity should correspond to the plasma itself, and based on that you can defined which venoms resulted in pro-coagulant or anti-coagulant. In this sense, I would suggest to change the x axis as fold in change, to show the data.
Methodology:
Why these particular Malaysian and Sri Lankan venoms were chosen for the study?
Do you know the type of antivenoms used in the study? Are all of them the whole IgG molecule?
Minor revisions:
1) Line 150 the concentrations of NH4SCN should be expressed in molarity (M) instead of moles.
2) Line 197: is found instead of fund.
3) Lines 300 and 304: in vitro and in vivo should be in italic
Author Response
Reviewer 1
The manuscript describes the possible cross-reactivity of Thai antivenoms against Asian viper venoms, specifically Malaysian and Sri Lankan venoms, which are not included in the immunization mixture to produce the antivenoms tested here. Overall, I found the manuscript interesting since antivenom availability is a valid concern and crisis particularly in parts of Asia. However, I have some questions and want to address some comments that, in my opinion, should be taken into consideration to improve the quality of the manuscript.
We thank the reviewer for their time and consideration of our manuscript. Below we respond in detail to their specific comments.
Introduction:
In the sentence starting in line 83, is addressed the lack of information regarding the binding capability of the antivenoms towards the toxins found in venoms not used in the immunization mixture. Even though is really well explained in the discussion, it should be mentioned also in the introduction that the immune recognition of these toxins does not necessarily mean effective neutralization.
Response: We agree with the reviewer comment. We have now made this point clear in the paragraph of the introduction mentioned by the reviewer.
Results and discussion:
In the immunological cross-reactivity determination through EPT ELISA, how it was established the cut-off value of the assay to consider a binding as significant?
Response: We specify a ‘cut-off’ value as indicative of substantial binding activity of each antivenom. The objective of this experiment was to investigate the trend of binding of each antivenom under increasing dilutions to each venom. Thus, in Figure 1 we display the binding profiles in full, and in Table 1 we selected a single discriminatory dilution to facilitate discussion of the results. This dilution was chosen for exactly that reason – it is the most discriminatory in terms of comparisons across the antivenoms studied.
In the immunological cross-reactivity visualized by immunoblotting with the Southeast Asian pit viper venoms, the protein profiles from the SDS-PAGE gels in figure A differ a lot from the immunoblotting results. You should displayed the results using the same molecular weight markers for both figures, in order to facilitate the comparison.
Response: We thank the reviewer for spotting this error. We already rearranged this figure as suggestion.
Regarding the neutralization of the coagulant activity, I have some concerns:
- a) You should add a control of antivenom only to assured that is not interacting with the plasma.
Response: A control of antivenom only was performed in these experiments and we have mentioned this in the result section. However, these data were not presented in the figures as we were concerned that this could make the data interpretation confusing for the reader. To address the reviewer concern though, the effect of antivenom on clotting activity without the addition of venom is now presented as supplementary data in this resubmission.
- b) The plasma employed in the assays, was rich or poor of platelets? Based on this information, in the discussion you should mention or not the platelets.
Response: We did centrifuge the plasma used in coagulation assay. Therefore the plasma we used in this assay is platelet poor plasma, and we have now specified this in the methods.
- c) Did you establish a minimal coagulation dose? I think this should be important since it could be the reason why you did not see neutralization of the activity displayed by hypnale, compared to the literature. Moreover, this would assure a better understanding of the neutralizing activity of the antivenoms tested here.
Response: Although the suggestion by the reviewer is sound, we preferred not to undertake minimum coagulation doses as part of our experimental strategy, instead directly comparing on a ng to ng basis the coagulant activity of each venom and thereafter neutralization by antivenoms using a dose equivalent to that recommended by the manufacturer.
- d) Why did you incubate the venoms with the antivenoms 10 min at room temperature and not as suggested by WHO when testing the efficacy of commercial antivenoms, which is 30 min at 37ºC?
Response: Our previous study (Reference number:5) demonstrated that a 10 min incubation was sufficient to allow neutralization activity between venoms and antivenom in this coagulation assay. Given that the WHO recommendation of co-incubation is completely artificial in the context of antivenom neutralization (and also doesn’t refer to this assay), any attempt to assess neutralization with reduced incubation times is beneficial.
- e) In addition, the way the results are expressed can be confusing since 100% of coagulant activity should correspond to the plasma itself, and based on that you can defined which venoms resulted in pro-coagulant or anti-coagulant. In this sense, I would suggest to change the x axis as fold in change, to show the data.
Response: This is a fair and logical point. We have modified this figure to now normalize “100% clotting” to the plasma control, with venoms that are procoagulant showing increased clotting activity (the majority), and those showing anticoagulant activity (T. fucatus) reduced clotting activity compared with the plasma control.
Methodology:
Why these particular Malaysian and Sri Lankan venoms were chosen for the study?
Response: Malaysia and Sri Lanka do not have their own antivenom production facility and rely on imported antivenom for their supplies. Information of the binding capability of the antivenoms to toxin in some venoms are unknown, especially for species which are uniquely endemic to those countries. For example, in Malaysia, all antivenoms for medically important species are purchased from Thai Red Cross. The Green Pit Viper antivenom and Hemo Polyvalent antivenom were produced using Trimeresurus albolabris, a species which is not found in Malaysia but has shown to have cross-reactivity to other species in genus Trimeresurus. There is considerable evidence of geographical variation in venom activity, and thus we hypothesized that geographical variation between Thai and Malaysian venoms could impact upon antivenom interactions. Thus, we sourced appropriate venoms available to us to study this phenomenon.
Do you know the type of antivenoms used in the study? Are all of them the whole IgG molecule?
Response: All of the studied antivenoms are F(ab')2 – we have added this information to the methods section of the manuscript.
Minor revisions:
1) Line 150 the concentrations of NH4SCN should be expressed in molarity (M) instead of moles.
Response: Amended as requested. We have deleted ‘moles’ and change it to ‘M’
2) Line 197: is found instead of fund.
Response: Amended as requested. We have deleted ‘fund’ and change it to ‘found’.
3) Lines 300 and 304: in vitro and in vivo should be in italic
Response: Amended as requested. We have changed both terms into italic.

Reviewer 2 Report
The authors performed two in vitro assays and an immunoblot to compare the binding capabilities of four major antivenoms against venom from eight species of medically important snakes. They then used an additional in vitro assay to compare the coagulopathic neutralizing activity of the antivenoms against the venoms administered to bovine plasma. Their work provides a thorough preclinical assessment of several common antivenoms available on the market in southeast Asia. Notably, their results reveal that one antivenom (HPAV) has broad-range effectiveness across, which has important implications for clinical use. Overall, it’s a solid study, well-written and presented, and the results are useful for downstream investigations aimed at further investigating these antivenoms in vivo. Below, I list my suggestions for improvement.
Major comments:
Is there a reason why IC50 was not used as a standard measurement for assessing the inhibition strength of each antivenom against the venoms for both the cross-reactivity and avidity tests? It’s not clear why comparisons were made between the 1:2500 dilutions instead. Since the authors use Prism for their statistical analyses of coagulopathic data, Prism can also be used to generate IC50s. Also, statistical comparisons would be helpful here.
Line comments:
Figure 3 legend – please explain what “M” represents in the first gel of panel A
L205 – why are Southeast Asian and Sri Lankan species venom immunoblots separated? The previous sections compare all 8 together. It’s easier to compare the data when they’re together.
L281 – “only” seems to be a typo here, perhaps another word was intended?
L510 – it’s unclear what was statistically analyzed. Please elaborate on this.
Author Response
Reviewer 2
The authors performed two in vitro assays and an immunoblot to compare the binding capabilities of four major antivenoms against venom from eight species of medically important snakes. They then used an additional in vitro assay to compare the coagulopathic neutralizing activity of the antivenoms against the venoms administered to bovine plasma. Their work provides a thorough preclinical assessment of several common antivenoms available on the market in southeast Asia. Notably, their results reveal that one antivenom (HPAV) has broad-range effectiveness across, which has important implications for clinical use. Overall, it’s a solid study, well-written and presented, and the results are useful for downstream investigations aimed at further investigating these antivenoms in vivo. Below, I list my suggestions for improvement.
We thank the reviewer for their kind words relating to our manuscript, and for their time spent providing these helpful comments, which we address one-by-one below.
Major comments:
Is there a reason why IC50 was not used as a standard measurement for assessing the inhibition strength of each antivenom against the venoms for both the cross-reactivity and avidity tests? It’s not clear why comparisons were made between the 1:2500 dilutions instead. Since the authors use Prism for their statistical analyses of coagulopathic data, Prism can also be used to generate IC50s. Also, statistical comparisons would be helpful here.
Response: We feel that an IC50 is not suitable to be used as a comparative parameter as both immunological assays do not assess the inhibition of antivenom binding towards venom or vice versa. Thus, instead we preferred to display the antivenom binding curves and use a single direct measurement based on the most discriminatory dilution factor for discussing our ELISA results.
Line comments:
Figure 3 legend – please explain what “M” represents in the first gel of panel A
Response: M represent lane that was loaded with a protein marker. We have added the description for M to the figure legend.
L205 – why are Southeast Asian and Sri Lankan species venom immunoblots separated? The previous sections compare all 8 together. It’s easier to compare the data when they’re together.
Response: We agree with the reviewer that the data will be easier compare when all immunoblots are placed together and so have merged the two immunoblotting figures together in the revised manuscript, as suggested by the reviewer.
L281 – “only” seems to be a typo here, perhaps another word was intended?
Response: Unfortunately, it was unclear to us exactly what this referred to, though we have checked the use of ‘only’ throughout.
L510 – it’s unclear what was statistically analyzed. Please elaborate on this.
Response: We have now added some detail to section 4.2.4 of the methods to make it clear that the statistical analyses were performed on the data resulting from the coagulation assay.

Round 2
Reviewer 1 Report
I consider that, after the revisions and corrections done by the authors, the manuscript is suitable to be published